# Determination of melon seed physical parameters and calibration of discrete element simulation parameters

Yuan Wan[1], Bowen Zhang[1], Qinghui Lai[2], Yan Gong[3], Yu Qingxu[3]*, Xiao Chen[3]

**1** Faculty of Modern Agriculture Engineering, Kunming University of Science and Technology, Kunming, Yunnan Province, China, **2** School of Energy and Environment Science, Yunnan Normal University, Kunming, Yunnan Province, China, **3** Nanjing Institute of Agricultural Mechanization, Ministry of Agriculture and Rural Areas, Nanjing, Jiangsu Province, China

* yqx5205@163.com

## Abstract

To improve the accuracy of the Hami melon discrete element model, the parameters of the Hami melon seed discrete element model were calibrated by combining practical experiments and simulation tests. The basic physical parameters of Hami melon seeds were obtained through physical experiments, including triaxial size, 100-grain mass, moisture content, density, Poisson's ratio, Young's modulus, shear modulus, angle of repose, suspension speed and various contact parameters. Taking the repose angle of seed simulation as an index, the parameters of each simulation model were significantly screened by the Plackett-Burman test. The results showed that the recovery coefficient, static friction coefficient and rolling friction coefficient of Hami melon seeds had significant effects on repose angle. Based on the steepest climbing test and quadratic regression orthogonal rotation combination test, it was determined that the significant order of the influence of various contact parameters on the angle of repose was static friction coefficient, collision recovery coefficient, and rolling friction coefficient. The optimal parameter combination was obtained through the mathematical regression model between the angle of repose and various contact parameters, namely, the collision recovery coefficient of Hami melon seeds was 0.518, the static friction coefficient of Hami melon seeds was 0.585 and the rolling friction coefficient of Hami melon seeds was 0.337. Under this condition, three static seed-dropping experiments and dynamic rolling accumulation experiments were carried out. The average simulated angle of repose was 31.93˚, and the relative error with the actual value was only 1.71%. The average simulated rolling accumulation angle was 51.98˚, and the relative error with the actual value was only 1.92%.

## 1. Introduction

Melons are widely commercialized in China, enjoying a high reputation and popularity in the international market. They have a broad planting area in China, primarily produced in the

**Funding:** This paper was supported by the National Natural Science Foundation of China (Grant No.31960366).

**Competing interests:** The authors have declared that no competing interests exist.

Hami region of Xinjiang Uygur Autonomous Region. In addition to being consumed as fresh produce, cantaloupes can be utilized in food processing, and melon seeds also have medicinal applications, showcasing both culinary and medicinal value and presenting promising prospects for development [1, 2].

Cantaloupe planting plays a critical role in improving cantaloupe production efficiency, and mechanized precision seeding is paramount [3] seed-metering device is the primary component of the seeding machine, and its strengths and weaknesses critically impact the machine's performance. The force acting on seeds during the seeding process is intricate, and employing the discrete element method [4–6] enables the study and analysis of seed motion and force features, which optimizes the parameters of the seed-metering device. To improve the accuracy of the discrete element simulation experiment in the discrete element software, it is necessary to accurately establish the discrete element model of the material and accurately define the physical parameters of the simulation model [7–9]. The material physical parameters mainly include intrinsic parameters and contact parameters [10], the intrinsic parameters are the object's own characteristic parameters (such as Poisson's ratio, Young's modulus and Density, etc.); the contact parameters between particles or between particles and geometries include Collision Recovery Coefficients, Static Sriction Factors and Rolling Friction Factors, which are the physical parameters of the contact between the two objects.

No research results are currently available on the physical parameters of cantaloupe seeds. Nonetheless, discrete element models have been created and simulation parameters calibrated for wheat, corn, fodder, and potato crops. COETZEE et al [11] calibrated the friction coefficient and Young's modulus of a discrete element model of corn using shear and compression experiments, respectively, which were validated by silo unloading and bucket loading experiments. Miyamoto et al. [12] calibrated the discrete meta-parameters of rice using angle of repose tests, which were subsequently experimentally verified with the assistance of a sorting machine. Yu et al [13] calibrated the contact parameters of the discrete element model for Panax ginseng seeds using the relative error between the measured and simulated stacking angles as the index. Ma et al [14] took the relative error of the angle of repose of the alfalfa and the relative error of the stacking angle as the test indexes and carried out multi-objective optimization calculation with the help of the genetic algorithm to improve the accuracy of the discrete element method in alfalfa seed discharger research. Shi et al [15] simulated the flow characteristics of the caraway seeds based on the Hertz-Mindlin model and calibrated the rolling friction coefficient of caraway seeds by simulation approximation prediction method. Wu et al [16] measured the angle of repose of the *Radix Peucedani* seeds by the elevated hollow cylinder stacking test method and combined with image processing techniques, and calibrated the physical parameters of *Radix Peucedani* seeds by the Plackett-Burman screening test, response surface analysis of variance, and optimization of regression equations. Wang et al [17] used two contact materials (Plexiglas plate and Aluminum cylinder) to conduct simulation tests on maize seed stacking angle, set up two binary regression equations with independent variables as static interseed friction coefficient and rolling interseed friction coefficient, and solved them numerically with the actual measurement of seed stacking angle as a known target value to find the maize interseed friction coefficient in EDEM.

In this paper, the physical test method was utilized to derive various parameters for cantaloupe seeds, including the triaxial dimensions, thousand-seed mass, water content, density, Young's modulus, Poisson's ratio, angle of repose, as well as collision recovery and static friction coefficients between the seeds and various contact materials. The physical test values established the foundation for selecting the simulation parameters [18]. The steepest climb test and quadratic regression orthogonal rotation combination test were executed to calibrate the discrete element simulation parameters of cantaloupe seeds. The seed simulation angle of

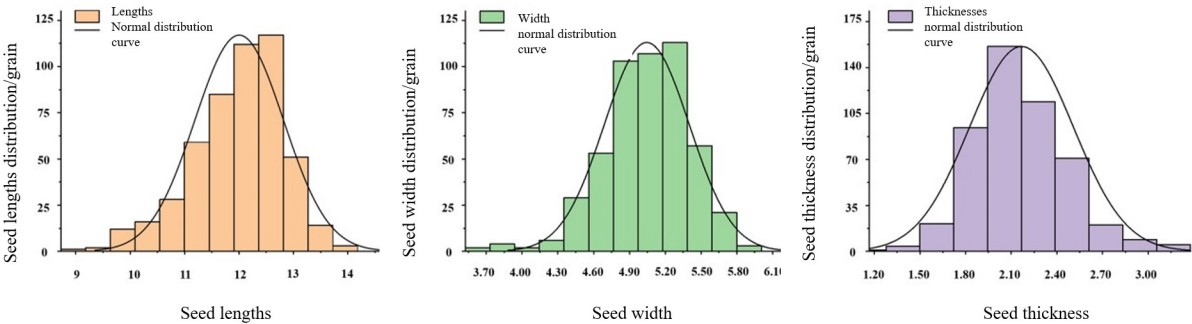

**Fig 1. Triaxial size distribution of Hami melon seeds.**

repose was utilized as the response value. Finally, we verified the reliability of the seed model parameters by comparing the seed drop test with the roller test. This provided us with theoretical parameters for the design and optimization of subsequent cantaloupe seed dischargers.

## 2. Materials and method

### 2.1 Determination of intrinsic parameters

Cantaloupe seeds are considered bulk materials, with their physical characteristics including triaxial dimensions, moisture content, 100-grain weight, density, Poisson's ratio, Young's modulus, and shear modulus. The experiment was carried out within the precincts of the Agricultural Materials Laboratory at Kunming University of Science and Technology. The subject of the investigation was the "Royal Princess 1028" model of high-grade cantaloupe seeds, which had been meticulously chosen and selectively cultivated by the Research Center of Xinjiang Academy of Agricultural Sciences.

**2.1.1 Three-axis size.** We selected 500 cantaloupe seeds randomly and measured their length, width, and thickness using highly accurate digital calipers with a precision of 0.01 mm. The resulting measurements, as shown in Fig 1, closely followed a normal distribution pattern.

**2.1.2 Hundred grain weight, moisture content and density.** The moisture content of the seeds was determined using the drying and weight reduction methods. Ten randomly selected cantaloupe seeds were analyzed using an MA-150 infrared moisture meter. Subsequently, 500 cantaloupe seeds were randomly selected and evenly divided into five groups. The 100-grain weight was measured using the LQ-C50002 model electronic balance with an accuracy of 0.01 g, while the density was determined using the gravimetric flask method. The measured values of the 100-grain weight, water content, and density of cantaloupe seeds are presented in Table 1.

**2.1.3 Poisson's ratio and shear modeling.** Young's modulus, shear modulus, and Poisson's ratio are intrinsic mechanical properties of materials [19] that describe the relationship between stresses and strains in materials. Measuring shear modulus in small, flat cantaloupe seeds can be challenging through physical tests. However, the Young's modulus and Poisson's

**Table 1. Physical property parameters of cantaloupe seeds.**

| Physical property parameter | Max | Min | Average |
|---|---|---|---|
| Moisture Content /% | 3.51 | 2.69 | 2.987 |
| Hundred-Grain Weight /g | 5.31 | 5.16 | 5.23 |
| Densities /(kg·m$^{-3}$) | 930 | 883 | 907 |

ratio can be determined through physical and mechanical tests. Shear modulus can be calculated using formula (1), and their relationship is expressed in Eq (1):

$$G = \frac{E}{2(1 + \lambda)} \tag{1}$$

Where: $G$—shear modulus of cantaloupe seeds, pa;
$E$—Young's modulus of cantaloupe seeds, pa;
$\lambda$—Poisson's ratio of cantaloupe seeds.

Poisson's ratio is an elastic constant that signifies the transverse deformation of the seed. It represents the ratio of positive transverse strain to positive axial strain when the seed undergoes unidirectional tensile or compressive stress. This ratio is calculated as shown in Eq (2).

$$\lambda = \frac{\varepsilon'}{\varepsilon} = \frac{\Delta w / w}{\Delta h / h} \tag{2}$$

Where: $\varepsilon'$—transverse positive strain of cantaloupe seeds;
$\varepsilon$—axial positive strain of cantaloupe seeds;
$\Delta w$—transverse deformation of cantaloupe seeds, mm;
$w$—width of cantaloupe seeds, mm;
$\Delta h$—axial deformation of cantaloupe seeds, mm;
$h$—thickness of cantaloupe seeds, mm.

The Young's modulus is used to measure a material's ability to resist elastic deformation, and it can be determined through tensile or compression tests. In cases where clamping seeds for tensile testing is challenging, compression tests are typically preferred. The mechanical properties of seeds under compression provide insights into their mechanical behavior when subjected to pressure. Moreover, these properties are essential for designing seed dischargers, assessing seed crushing forces, and conducting related studies [20]. The formula for calculating the Young's modulus is given by Eq (3).

$$E = \frac{\sigma}{\varepsilon} = \frac{F / A}{\Delta h / h} \tag{3}$$

Where: $\sigma$—axial positive stress of cantaloupe seeds, pa;
$F$—axial force of cantaloupe seeds, N;
$A$—cross-sectional area of cantaloupe seeds, mm$^2$.

In this paper, we conducted compression mechanical tests on cantaloupe seeds using the TA.XTPlus professional food physical property analyzer developed by Stable Micro Systems Company, UK, with an accuracy of 0.0002%. The mechanical properties of the seeds are illustrated in Fig 2.

We selected ten seeds with triaxial dimensions close to the average values for the compression test. These seeds were placed sequentially on the pressurized table of the texture instrument, and a rigid indenter from the SMS P/36R model was used for pressure loading. The indenter's loading speed was set to 0.1 mm/s, and the initial force applied was 0.2 N. This setup effectively demonstrates the macroscopic rupture of cantaloupe seeds under squeezing pressure. We then collected and processed the data from the test, resulting in the compression force-displacement curve depicted in Fig 3.

From the compression force-displacement curve in Fig 3, it's evident that the displacement falls within the 0 to 0.3 mm range. The relationship between compression force and displacement is predominantly linear, indicating behavior within the realm of linear elasticity, with stress and strain conforming to Hooke's law. By extracting compression force data at a

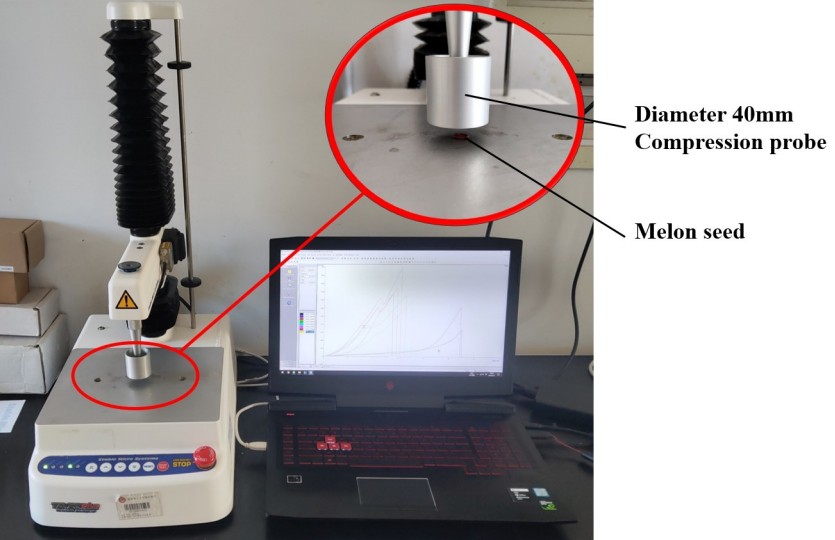

**Fig 2. Experiment on mechanical properties of Hami melon seeds.**

displacement of 0.3 mm, we obtained an average axial force value ($F$) of 4.41 N, an average seed cross-sectional area ($A$) of 48.125 mm$^2$, and an average seed thickness ($h$) of 2.2 mm. Using Eq (3), we calculated an average Young's modulus ($E$) of 0.67 MPa.

Due to the small size and hard skin of cantaloupe seeds, measuring their axial deformation is challenging as it is minimal during rupture, making it difficult to measure with calipers. As shown in the compression force-displacement curve in Fig 3, the average axial displacement during seed rupture in the thickness direction is 0.85 mm. To address this, we selected ten seeds with triaxial dimensions close to the average and conducted a compression test with the

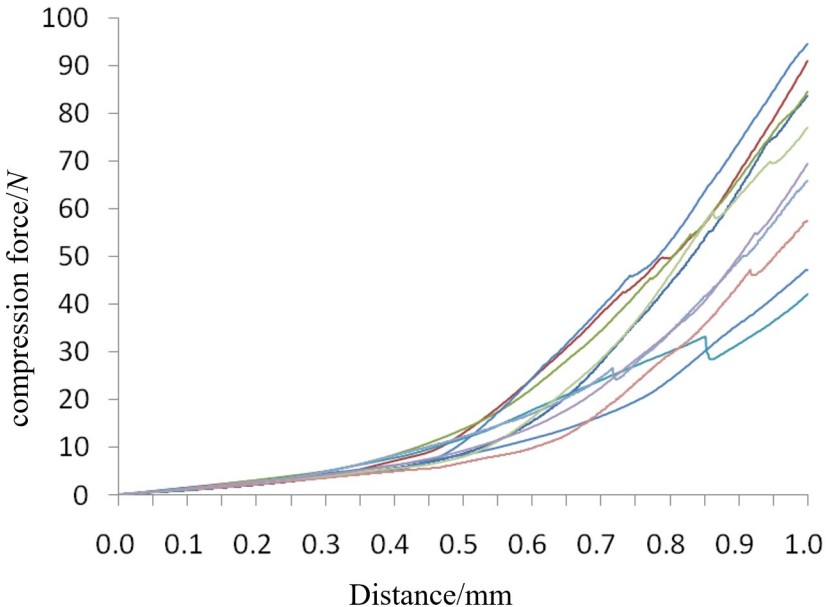

**Fig 3. Compression force-displacement curve.**

indenter's fixed stroke set to 0.85 mm, using the same operating parameters. The average transverse deformation of the seeds after compression was measured at 0.776 mm using digital calipers, with an average seed width ($w$) of 5.07 mm. Calculating the average Poisson's ratio ($\lambda$) of the seeds using formula (2), we obtained a value of 0.4. Furthermore, the average shear modulus ($G$) of the seeds was calculated as 0.24 MPa using formula (1).

**2.1.4 Measurement of contact parameters and simulation calibration.** The pre-designed components of the seed dispenser are 3D printed using ABS plastic. In this paper, we determine the contact parameters between cantaloupe seeds and between cantaloupe seeds and ABS plastic through physical tests and calibrate them using simulation tests. These contact parameters encompass the collision recovery coefficient, static friction coefficient, and rolling friction coefficient. Cantaloupe seeds are small and flat, making the direct measurement of real contact parameters challenging. These values exhibit significant variation, and some parameters are difficult to measure directly. Additionally, there may be discrepancies between the discrete element model of the seeds and their actual dimensions, resulting in errors between the simulated contact parameters and the actual ones. To enhance the reliability of the discrete element simulation tests, this paper employs a combination of bench tests and simulation tests. We directly measure the collision recovery coefficient and static friction coefficient between seeds and ABS plastic through bench tests and calibrate the collision recovery coefficient, static friction coefficient, and rolling friction coefficient among the seeds, as well as the rolling friction coefficient between seeds and ABS plastic through simulation tests.

## 2.2 Bench testing

**2.2.1 Collision recovery coefficient between seed and ABS plate.** In the mechanical sowing process, seeds are subject to mutual collisions and interactions with the sowing machinery. Investigating the collision recovery coefficient of seeds holds significant importance [21]. The Seed collision recovery coefficient is a fundamental granular property required for building a discrete element simulation. The accuracy of the simulation output results depends on the accuracy of the input parameters of the DEM [22, 23]. This coefficient quantifies an object's ability to recover after experiencing deformation due to a collision. The schematic diagram for the collision recovery coefficient test is depicted in Fig 4.

The collision recovery coefficient, defined as the ratio of the normal partial velocity of the seed after the collision to the normal partial velocity before the collision, is a crucial parameter [24]. The formula is as follows:

$$C_r = \frac{U_n}{V_n} = \frac{U \bullet \sin\beta}{V \bullet \sin\alpha} \tag{4}$$

$$V = \sqrt{2gH_0} \tag{5}$$

$$\begin{cases} U = \sqrt{U_x^2 + U_y^2} \\ L_1 = U_x t_1 \\ H_1 = U_y t_1 + \frac{1}{2} g t_1^2 \end{cases} \tag{6}$$

$$arctan\frac{Uy}{Ux} + \alpha + \beta = 90° \tag{7}$$

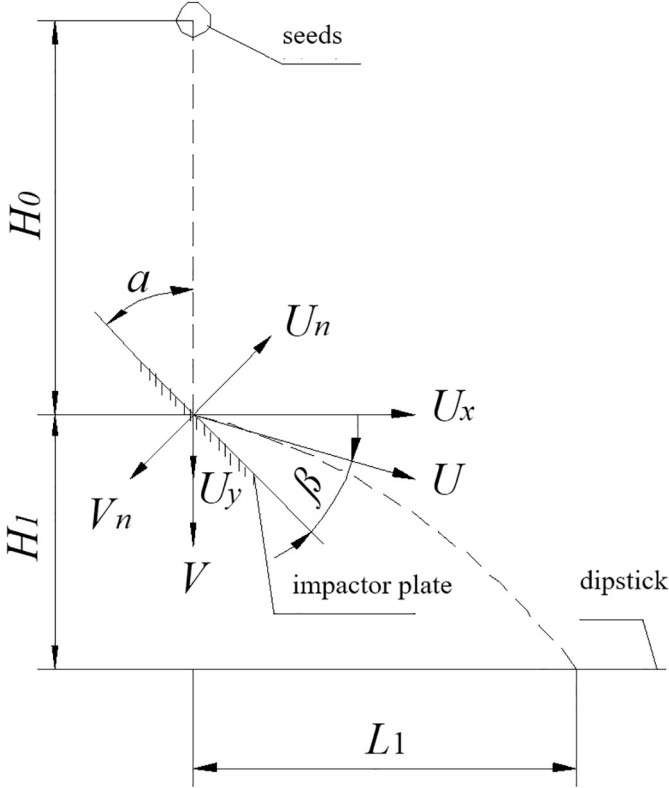

**Fig 4. Experimental schematic diagram of collision recovery coefficient.**

Eq (8) is obtained by organizing Eqs (4), (5), (6) and (7):

$$C_r = \frac{\sqrt{L_1^2 + \left(H_1 - \frac{1}{2}gt_1^2\right)^2} \bullet sin\left(90\text{-}\alpha\text{-}arctan\frac{H_1 - \frac{1}{2}gt_1^2}{L_1}\right)}{t_1 \bullet \sqrt{2gH_0} \bullet sin\alpha} \tag{8}$$

Where: $C_r$—collision recovery coefficient;
$V$—seed pre-collision velocity, m/s;
$U$—seed velocity after collision, m/s;
$V_n$—Normal velocity of the seeds before the collision, m/s;
$U_n$—normal velocity of the seeds after collision, m/s;
$U_x$—x-axis component of velocity after seed collision, m/s;
$U_y$—y-axis component of velocity after seed collision, m/s;
$H_0$—free-fall height of the seed before collision, m;
$H_1$—the falling height of the seed after the collision, m;
$L_1$—horizontal displacement of the seed after collision, m;
$t_1$—the time of the seed's planar motion after collision, s;
$\alpha$—angle of the collision plate, °.

In accordance with the physical definition of the collision recovery coefficient in energetics, this paper presents the design of the cantaloupe seed collision recovery coefficient determination device, as illustrated in Fig 5. The impact plate is set at an angle of $\alpha = 45°$, and ABS plastic

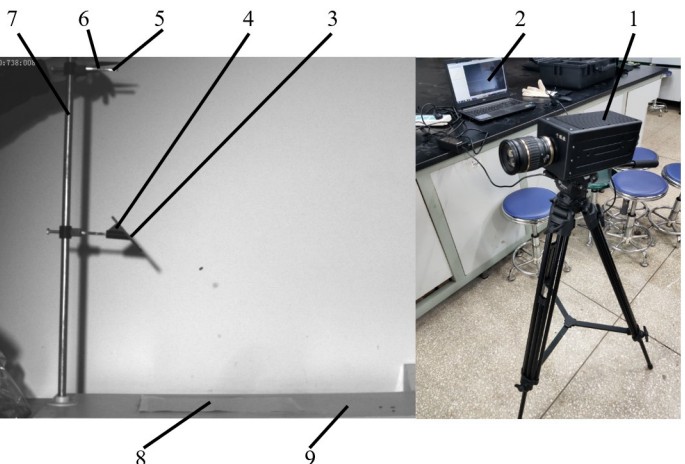

**Fig 5. Measuring device of collision recovery coefficient.**

is used as the material to be tested on the impact plate. The test seeds start from a state of rest with an initial velocity of zero, dropping freely from a height of $H_0 = 400$ mm through a designated drop hole. Upon collision with the ABS plastic plate, the seeds follow a parabolic trajectory.

In this study, we utilized a high-speed camera of the Thousand Eyes Wolf 5F01 type with specific parameter settings. We captured video files of falling cantaloupe seeds in areas where seed aggregation exceeded 90%. We recorded data for various variables, including the seed's post-collision falling height ($H_1$), the average duration of its parabolic motion ($t_1$), and the horizontal displacement of the seed on the collector plate after collision ($L_1$). Using the formula (8), we then calculated and analyzed the collision recovery coefficients for Hami melon seeds interacting with ABS plate. The average result was 0.51.

**2.2.2 Coefficient of static friction between seed and ABS plate.** The coefficient of friction is influenced by several factors, for instance, the material type and the roughness of the contact surface. Determining the friction coefficient for different bulk materials holds significant importance in guiding engineering applications. It serves as crucial foundational data for the practical design of bulk material transportation and processing equipment. Depending on the specific size parameters and physical and chemical properties of bulk materials, various methods, and equipment, such as the tiltmeter, turntable tester, and single-fiber sliding friction coefficient tester, can be employed to test the friction coefficients of different bulk materials. The coefficient of static friction is closely related to the mobility of the seed within the discharger, the energy consumption of the discharger, and the mechanical wear between the discharger and the seed in seed discharger design [25]. In this study, we utilized the tiltmeter method to determine the static friction coefficient, as illustrated in Fig 6. When an object is placed on the inclined surface and remains stationary, the static friction coefficient can be calculated as:

$$Gsin\gamma = \mu Gcos\gamma \tag{9}$$

Where, $G$—seed weight, N;
$\gamma$—inclination angle of the inclined plane, ˚;
$\mu$—coefficient of static friction.

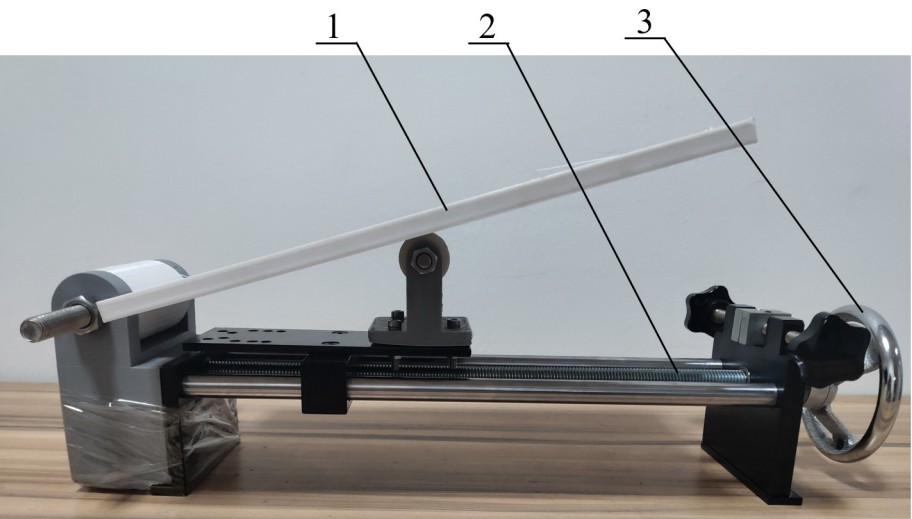

**Fig 6. Friction coefficient measuring device.**

Then there are:

$$\mu = tan\gamma \tag{10}$$

When the inclination of the inclined plane increases, the objects on the surface exhibit a tendency to slide. At this point, the inclination of the inclined plane ($\gamma$) corresponds to the static friction angle of the object on the inclined plane. The static friction angle of the object is measured using an angle ruler, and the static coefficient of friction for the object is determined through the calculation of formula (10).

During seeding research, 3D printing is utilized for seed dispenser fabrication due to its efficient and rapid processing of complex parts, which accelerates the research process and ABS is a commonly used 3D printing material. In this paper, we used a SWI-1 type homemade inclinometer to measure the static friction coefficient. The plate to be tested was replaced with an ABS plastic plate, and the test was repeated 10 times. The resulting average value of the static friction coefficient between the cantaloupe seed and the ABS plate was 0.56.

**2.2.3 Angle of repose determination.** The angle of repose reflects the internal friction characteristics and scattering performance of bulk materials. It is an inherent property of materials and is closely related to several influencing factors, such as contact materials, material shape, material size, material moisture content, and the environment in which the materials are stacked. By adjusting the factors that influence the angle of repose, the simulation can more accurately reflect real-world conditions, thereby increasing its reliability and accuracy. The angle of repose is typically determined using the injection method. The device used for measuring the angle of repose by the injection method is depicted in Fig 7a, comprising a funnel and a disk. During the test, cantaloupe seeds are placed into the funnel, and they flow into the disk under gravity, forming a cone. The angle at the base of the cone represents the angle of repose and is captured by a high-definition camera, as shown in Fig 7b.

To minimize errors caused by human measurement, this experiment utilized Matlab software to sequentially perform noise reduction, grayscale processing, and binarization on the acquired angle of repose images. This process obtained the seed stack boundary points, which were then used to generate the seed stack boundary curves. Finally, the least squares method is used to fit a straight line to the boundary points, as shown in Fig 8c. The horizontal and

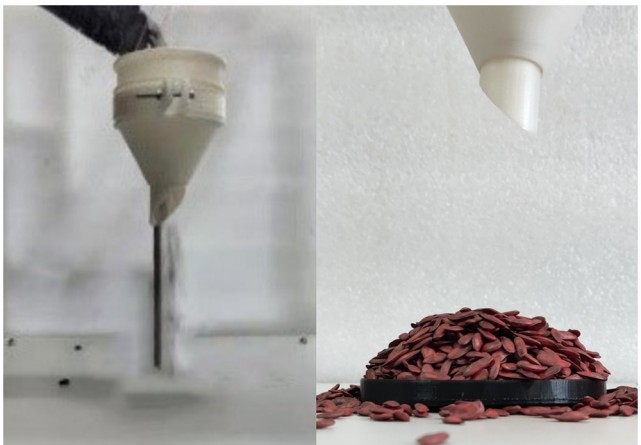

a. Seed-dropping test          b. Test results

**Fig 7. Measurement of angle of repose by injection method.**

vertical coordinates represent the pixel points of the image, without any practical dimensions (units). The slope of the fitted straight line corresponds to the tangent of the stacking angle that needs to be measured. The tests were repeated five times to obtain an average value, and the actual angle of repose of cantaloupe seeds was found to be 31.39°.

## 2.3 Simulation experiment

**2.3.1 Discrete meta-simulation modeling of seeds.**   Based on the three-axis dimensions and shapes of cantaloupe seeds, 3D solid modeling of these seeds was performed using Solid-Works software. The resulting model was saved as a.step file and subsequently imported into ANSYS software for mesh generation. It was then saved as a.msh file and further imported into EDEM software to create a multisphere polymerized particle model of the seeds, using the particle fast-filling function. This model is illustrated in Fig 9.

To ensure similarity of the seed appearance models while taking into account computational accuracy and speed, we randomly selected 300 spherical particles with diameters ranging from 0.1 to 1.19 mm to populate the seed-models.

**2.3.2 Angle of repose simulation test model.**   The 3D model of the funnel and the bottom plate was created in SolidWorks software, based on the parameters of the actual seed drop test. This model was then imported into EDEM, as depicted in Fig 10. Within the EDEM software, a simulation of the cantaloupe seed drop test was conducted. The test established a particle factory at the funnel's mouth, generating particles with dynamic random positions. A total of 1500 seed-models were generated at a rate of 600 seed-models per second, with data saved at 0.01-second intervals. To ensure a complete pile-up of seeds on the base plate, the total simulation time was set at 2.85 seconds, and the fixed time step represented 15% of the Rayleigh time.

In this paper, a simulation test was conducted by varying the rolling friction coefficient between cantaloupe seed species and the rolling friction coefficient between seeds and the ABS plate. This aimed to observe how changes in these coefficients affected the angle of repose. All other parameters used in the theoretical test measurements mentioned above were utilized in this experiment. Through an extensive series of one-factor simulation tests, the initial range of the rolling friction coefficient was determined, as detailed in Table 2.

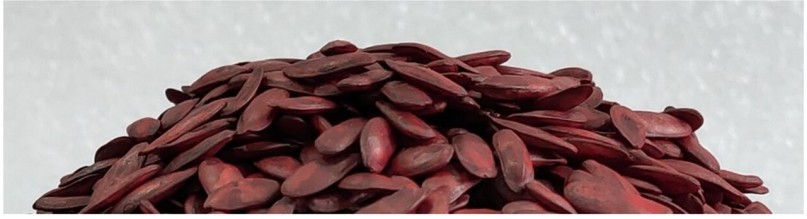

a. Original image

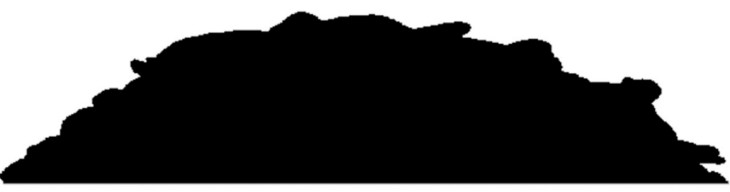

b. Binary image

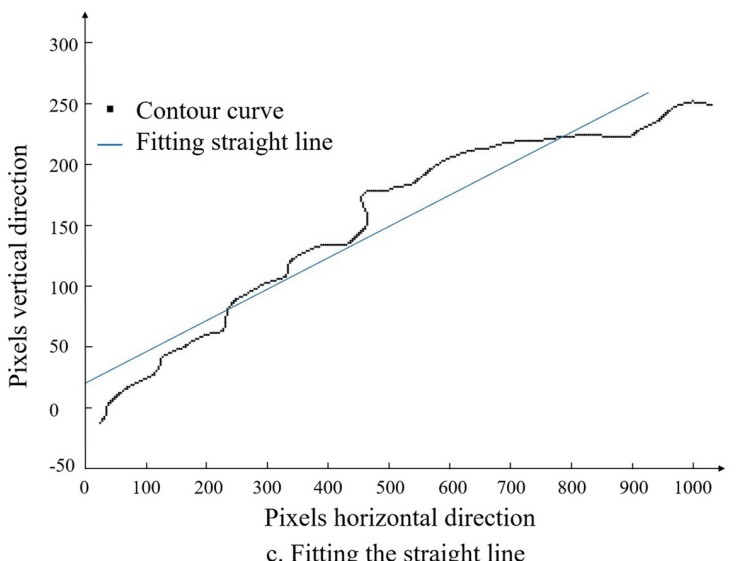

c. Fitting the straight line

**Fig 8. Image processing.**

## 2.4 Discrete element simulation parameter calibration

**2.4.1 Significance parameter determination.** To further determine the significant parameters affecting the angle of repose of the seeds, this paper uses it as an indicator and uses Design-Expert software to conduct the Plackett-Burman test to screen out the parameters that have a significant effect on the response values. The eight test parameters in Table 3 were assigned values according to the data above, and the high and low levels of the Plackett-Burman test were taken as the range of the test parameters. The design scheme and results of the Plackett-Burman test are shown in Table 4, and the angle of repose of the simulation was also obtained using the Matlab software.

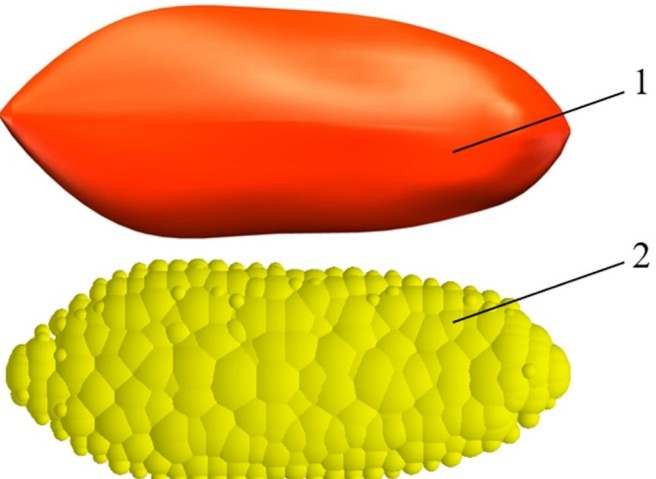

**Fig 9. 3D and DEM model of melon seed.**

P-tests were conducted on the results presented in Table 4 to determine the significance of each parameter on the angle of repose. The significant results are shown in Table 5. The P-values for parameters $D_s$ and $E_s$ are both less than 0.01, indicating that interspecific static friction coefficients and interspecific rolling friction coefficients have a highly significant impact on the angle of repose. The P-value for parameter $C_s$ is less than 0.05, signifying that interspecific collision recovery coefficients have a significant effect on the angle of repose. On the other hand, the P-values for the remaining parameters are greater than 0.05, suggesting that these parameters do not have a significant effect on the angle of repose.

**2.4.2 Steepest climb test.** Based on the results of the Plackett-Burman test, the steepest climb test was carried out for the three parameters that had a significant influence on the angle of repose. The relative error between the simulated angle of repose and the actual angle of

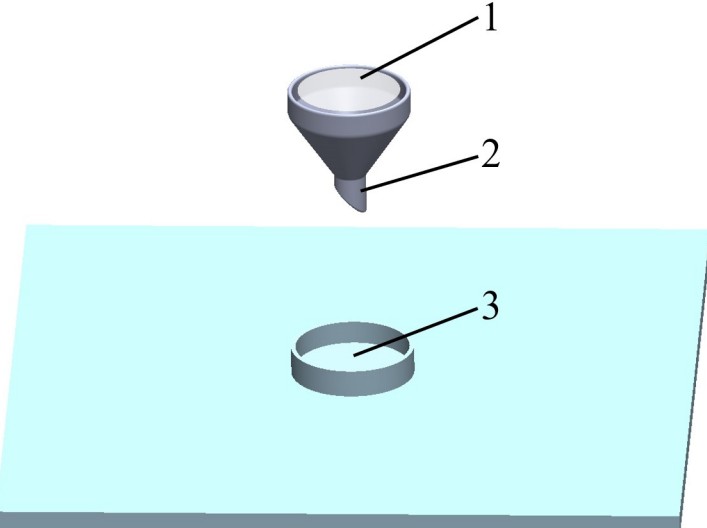

**Fig 10. Angle of repose simulation test model.**

**Table 2. Physical parameters of cantaloupe seeds.**

| Test parameters | retrieve a value | note |
|---|---|---|
| Poisson's ratio of Hami melon seeds | 0.4 | we have determined in this paper |
| Shear modulus of Hami melon seeds (MPa) | 0.24 | we have determined in this paper |
| Recovery coefficient of interspecific collision of Hami melon seeds | 0.39~0.63 | Simulation Calibration |
| Interspecific static friction coefficient of Hami melon seeds | 0.44~0.74 | Simulation Calibration |
| Inter-species rolling friction coefficient of Hami melon seeds | 0.2~0.5 | Simulation Calibration |
| Recovery coefficient of Hami melon seed -ABS plate collision | 0.51 | we have determined in this paper |
| The static friction coefficient of Hami melon seed -ABS plate | 0.56 | we have determined in this paper |
| Rolling friction coefficient of Hami melon seed -ABS plate | 0.18~0.38 | Simulation Calibration |

**Table 3. Table of test parameter ranges.**

| Test parameters | Low level(-1) | Hight level(+1) |
|---|---|---|
| Poisson's ratio of Hami melon seeds | 0.3 | 0.5 |
| Shear modulus of Hami melon seeds (MPa) | 0.12 | 0.36 |
| Recovery coefficient of interspecific collision of Hami melon seeds | 0.39 | 0.63 |
| Interspecific static friction coefficient of Hami melon seeds | 0.44 | 0.74 |
| Inter-species rolling friction coefficient of Hami melon seeds | 0.2 | 0.5 |
| Recovery coefficient of Hami melon seed -ABS plate collision | 0.41 | 0.61 |
| The static friction coefficient of Hami melon seed -ABS plate | 0.43 | 0.69 |
| Rolling friction coefficient of Hami melon seed -ABS plate | 0.18 | 0.38 |

repose was used as the test index to determine the optimal range of the test parameters. The design scheme and results of the steepest climb test are presented in Table 6.

**2.4.3 Quadratic regression orthogonal rotation combination test.** The results of the steepest climb test indicated that the relative error of the cantaloupe seed angle of repose

**Table 4. Plackett-Burman test scheme and results.**

| No. | experimental parameter | | | | | | | | Angle of repose θ/(°) |
|---|---|---|---|---|---|---|---|---|---|
| | $A_s$ | $B_s$ | $C_s$ | $D_s$ | $E_s$ | $F_s$ | $G_s$ | $H_s$ | |
| 1 | 0.5 | 0.36 | 0.39 | 0.74 | 0.5 | 0.58 | 0.45 | 0.18 | 27.23 |
| 2 | 0.3 | 0.36 | 0.63 | 0.44 | 0.5 | 0.58 | 0.71 | 0.18 | 36.73 |
| 3 | 0.5 | 0.12 | 0.63 | 0.74 | 0.2 | 0.58 | 0.71 | 0.38 | 37.69 |
| 4 | 0.3 | 0.36 | 0.39 | 0.74 | 0.5 | 0.38 | 0.71 | 0.38 | 39.31 |
| 5 | 0.3 | 0.12 | 0.63 | 0.44 | 0.5 | 0.58 | 0.45 | 0.38 | 36.02 |
| 6 | 0.3 | 0.12 | 0.39 | 0.74 | 0.2 | 0.58 | 0.71 | 0.18 | 38.33 |
| 7 | 0.5 | 0.12 | 0.39 | 0.44 | 0.5 | 0.38 | 0.71 | 0.38 | 33.09 |
| 8 | 0.5 | 0.36 | 0.39 | 0.44 | 0.2 | 0.58 | 0.45 | 0.38 | 27.11 |
| 9 | 0.5 | 0.36 | 0.63 | 0.44 | 0.2 | 0.38 | 0.71 | 0.18 | 29.55 |
| 10 | 0.3 | 0.36 | 0.63 | 0.74 | 0.2 | 0.38 | 0.45 | 0.38 | 35.43 |
| 11 | 0.5 | 0.12 | 0.63 | 0.74 | 0.5 | 0.38 | 0.45 | 0.18 | 42.44 |
| 12 | 0.3 | 0.12 | 0.39 | 0.74 | 0.2 | 0.38 | 0.45 | 0.18 | 27.35 |
| 13 | 0.4 | 0.24 | 0.51 | 0.59 | 0.35 | 0.48 | 0.58 | 0.28 | 33.44 |

**Table 5. Significance analysis of the parameters of the Plackett-Burman test.**

| Parameters | Degree of freedom | Sum of squares | F-value | P-value |
|---|---|---|---|---|
| $A_s$ | 1 | 3.06 | 1.72 | 0.281 |
| $B_s$ | 1 | 7.62 | 4.27 | 0.131 |
| $C_s$ | 1 | 19.87 | 11.14 | 0.044* |
| $D_s$ | 1 | 137.23 | 76.98 | 0.003** |
| $E_s$ | 1 | 71.83 | 40.29 | 0.008** |
| $F_s$ | 1 | 2.94 | 1.65 | 0.289 |
| $G_s$ | 1 | 6.93 | 3.89 | 0.143 |
| $H_s$ | 1 | 0.74 | 0.42 | 0.565 |

Note:

**Indicates that the impact is extremely significant ($P<0.01$), and

* indicates that the impact is significant ($P<0.05$).

The same as below.

**Table 6. Steepest climb test design options and results.**

| No. | Restitution coefficient of seed collision in Melon $X_1$ | The static friction coefficient of melon seeds $X_2$ | Rolling friction coefficient between melon seeds $X_3$ | Repose angle θ/ (°) | Relative error/ % |
|---|---|---|---|---|---|
| 1 | 0.39 | 0.44 | 0.20 | 25.32 | 19.34 |
| 2 | 0.43 | 0.49 | 0.25 | 27.54 | 12.27 |
| 3 | 0.47 | 0.54 | 0.30 | 30.22 | 3.73 |
| 4 | 0.51 | 0.59 | 0.35 | 32.17 | 2.48 |
| 5 | 0.55 | 0.64 | 0.40 | 33.23 | 5.86 |
| 6 | 0.59 | 0.69 | 0.45 | 33.69 | 7.33 |
| 7 | 0.63 | 0.74 | 0.50 | 36.61 | 16.63 |

initially decreased and then increased. The smallest relative error was observed in the vicinity of Experiment No. 4. This suggests that the optimal range interval is near Experiment No. 4. To further explore the optimal combinations of collision recovery coefficients, coefficients of static friction, and coefficients of rolling friction among cantaloupe seeds in EDEM simulation experiments, a three-factor quadratic regression study with an orthogonal rotational combination test was conducted. Three, four, and five groups of test factors were selected as the upper limit, the neutral level, and the lower limit for the quadratic regression orthogonal rotational combination test. The coding table of factors for the simulation experiment is presented in Table 7. In Table 7, $X_1$, $X_2$, and $X_3$ represent the collision recovery coefficient, static friction coefficient, and rolling friction coefficient between cantaloupe seeds, respectively. The design

**Table 7. Code table of simulation test factors.**

| Code | Factors | | |
|---|---|---|---|
| | $X_1$ | $X_2$ | $X_3$ |
| -1.682 | 0.47 | 0.54 | 0.30 |
| -1 | 0.49 | 0.56 | 0.32 |
| 0 | 0.51 | 0.59 | 0.35 |
| 1 | 0.53 | 0.62 | 0.38 |
| 1.682 | 0.55 | 0.64 | 0.40 |

**Table 8. Test scheme and results.**

| No. | Experimental factors | | | Y/% |
|---|---|---|---|---|
| | $X_1$ | $X_2$ | $X_3$ | |
| 1 | -1 | -1 | -1 | 5.32 |
| 2 | 1 | -1 | -1 | 0.88 |
| 3 | -1 | 1 | -1 | 8.47 |
| 4 | 1 | 1 | -1 | 5.61 |
| 5 | -1 | -1 | 1 | 3.42 |
| 6 | 1 | -1 | 1 | 0.69 |
| 7 | -1 | 1 | 1 | 6.13 |
| 8 | 1 | 1 | 1 | 5.81 |
| 9 | -1.682 | 0 | 0 | 5.47 |
| 10 | 1.682 | 0 | 0 | 1.49 |
| 11 | 0 | -1.682 | 0 | 0.93 |
| 12 | 0 | 1.682 | 0 | 5.77 |
| 13 | 0 | 0 | -1.682 | 5.51 |
| 14 | 0 | 0 | 1.682 | 3.73 |
| 15 | 0 | 0 | 0 | 2.48 |
| 16 | 0 | 0 | 0 | 1.87 |
| 17 | 0 | 0 | 0 | 1.81 |
| 18 | 0 | 0 | 0 | 2.09 |
| 19 | 0 | 0 | 0 | 1.27 |
| 20 | 0 | 0 | 0 | 0.93 |
| 21 | 0 | 0 | 0 | 2.34 |
| 22 | 0 | 0 | 0 | 1.74 |
| 23 | 0 | 0 | 0 | 1.21 |

scheme and results of the simulation experiments are listed in Table 8, and the experimental results are represented as the relative error, denoted as $Y$, between the simulated angle of repose and the actual angle of repose.

Design-Expert software was used to fit the multiple regression to the test data, and the regression equation for the relative error in the angle of repose $Y$ was obtained as follows:

$$Y = 1.74 - 1.25X_1 + 1.75X_2 - 0.53X_3 + 0.49X_1X_2 + 0.53X_1X_3$$
$$-0.0062X_2X_3 + 0.74X_1^2 + 0.69X_2^2 + 1.14X_3^2 \tag{11}$$

The significance test of the regression equation is shown in Table 9, and the fit of the model was highly significant ($P < 0.01$). The interaction term ($X_2X_3$) of static and rolling friction coefficients between cantaloupe seeds had $P > 0.1$, which was not significant for the relative error in the angle of repose, and the rest of the P-tests were significant, indicating that the test factors were not in a simple linear relationship to the response values, and there was a quadratic relationship. The loss of fit term $P = 0.2680$ is not significant, indicating that there are no other major factors affecting the index. The coefficient of determination of the regression equation $R^2 = 0.96$, indicating that the predicted values of the regression equation fit well with the actual values. The factors affecting the relative error of the angle of repose, from largest to smallest, were the interspecific static friction coefficient, the interspecific collision recovery coefficient, and the rolling friction coefficient of cantaloupe seeds.

**Table 9. Variance analysis of regression equation.**

| Source of variance | Sum of squares | Degree of freedom | F-value | P-value |
|---|---|---|---|---|
| Model | 107.34 | 9 | 35.27 | <0.0001** |
| $X_1$ | 21.27 | 1 | 62.91 | <0.001** |
| $X_2$ | 41.65 | 1 | 123.18 | <0.0001** |
| $X_3$ | 3.82 | 1 | 11.30 | 0.0051** |
| $X_1 X_2$ | 1.99 | 1 | 5.89 | 0.0306* |
| $X_1 X_3$ | 2.26 | 1 | 6.68 | 0.0227* |
| $X_2 X_3$ | 0.00 | 1 | 0.00 | 0.9762 |
| $X_1^2$ | 8.61 | 1 | 25.48 | 0.0002** |
| $X_2^2$ | 7.57 | 1 | 22.39 | 0.0004** |
| $X_3^2$ | 20.63 | 1 | 61.00 | <0.0001** |
| Residual | 4.40 | 13 | 1.58 | 0.2680 |
| Lack of fit | 2.19 | 5 | | |
| Error | 2.21 | 8 | | |
| The sum | 111.73 | 22 | | |

## 2.5 Determination of optimal parameters

The optimization module in Design-Expert software was employed to solve the regression equations, analyze the response surfaces, and optimize the regression model to minimize the relative error in the angle of repose for cantaloupe seeds. The objective and constraint equations are as follows:

$$\begin{cases} minY(X_1, X_2, X_3) \\ \text{s.t.} \begin{cases} 0.47 \le X_1 \le 0.55 \\ 0.54 \le X_2 \le 0.64 \\ 0.3 \le X_3 \le 0.4 \end{cases} \end{cases} \quad (12)$$

Several sets of optimal parameters were obtained, and in the end, the coefficient of collision recovery between cantaloupe seeds was chosen as 0.518, the static friction coefficient as 0.585, and the rolling friction coefficient as 0.337.

## 2.6 Verification test

To verify the reliability of the discrete element model of cantaloupe seeds, the mean values of the parameters identified above, along with the non-significant parameters, were used as the EDEM simulation parameters for the experimental validation of the static seed drop rest angle and the dynamic rolling stacking inclination. The validation results are shown in Fig 11.

The angle of repose of cantaloupe seeds was compared between the validation results of the static seed drop test and the actual test. In the actual test, the angle of repose was 31.39˚, while in the simulation test, it was 31.93˚, resulting in a 1.71% higher simulation result compared to the actual test. The validation results of the dynamic rolling stacking test were compared. The dynamic rolling stacking angle of cantaloupe seeds in the actual test was 51˚, while in the simulation test, it was 51.98˚, resulting in a 1.92% higher simulation result than the actual test. This indicates good accuracy of the discrete element model, providing a reliable basis for subsequent simulation studies of the seed discharger.

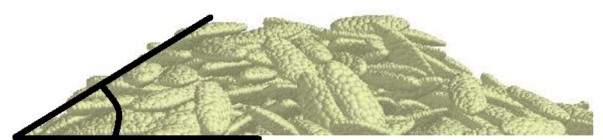

**Angle of repose of simulation test**

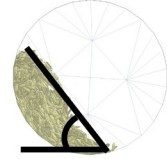

**Angle of rolling friction of simulation test**

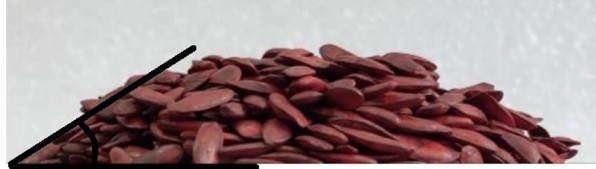

**Angle of repose of physical test**

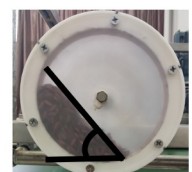

**Angle of rolling friction of physical test**

**Fig 11. Comparison of the angle of repose test of melon seeds.**

## 3. Results and conclusion

1. The fundamental physical parameters of cantaloupe seeds, such as triaxial size, hundred-grain mass, moisture content, density, Young's modulus, shear modulus, and Poisson's ratio, were determined through physical tests. The collision recovery coefficient and static friction coefficient between cantaloupe seeds and an ABS plate were found to be 0.51 and 0.56, respectively, by a Thousand Eyes Wolverine 5F01 high-speed video camera and SWI-1 inclinometer. The angle of repose of the seeds was determined to be 31.39° through a seed-dropping test conducted with Matlab software.

2. The physical parameters obtained from the physical tests served as the foundation for selecting simulation parameters. The Plackett-Burman test was conducted with the assistance of Design-Expert software to identify parameters significantly affecting the angle of repose for cantaloupe seeds. The key parameters were identified as the collision recovery coefficient, static coefficient of friction, and rolling coefficient of friction. Subsequently, the steepest-climbing test was performed to define the optimal range interval for these significant parameters.

3. By designing quadratic regression orthogonal rotational combination tests, we determined the optimal contact parameters between cantaloupe seeds in EDEM simulation experiments and assessed the impact of these parameters on the angle of repose. The optimal values for the coefficients of recovery from collision, static friction, and rolling friction between cantaloupe seeds were found to be 0.518, 0.585, and 0.337, respectively. The factors influencing the relative error in the angle of repose, from most to least significant, were the interspecies static friction coefficient, interspecies collision recovery coefficient, and rolling friction coefficient of cantaloupe seeds.

4. Simulation tests and bench tests were conducted to verify the static seed drop rest angle and dynamic rolling stacking inclination under the optimal parameter combinations. The results indicated that the error of the static seed drop rest angle for cantaloupe seeds was 1.71%, and the error of the dynamic rolling stacking inclination was 1.92%. These results closely matched the outcomes of actual experiments, confirming the reliability of the discrete element model and providing a theoretical foundation for subsequent simulation research on pneumatic suction seed dischargers.

## Supporting information

**S1 File. The data set of Fig 1.**
(XLSX)

**S2 File. The data set of Table 1.**
(XLSX)

**S3 File. The data set of Poissons ratio.**
(XLSX)

**S4 File. The data set of Fig 3.**
(XLSX)

**S5 File. The data set of bench testing.**
(XLSX)

## Acknowledgments

We express our gratitude to Qingxu Yu and Qinghui Lai for their support and constructive comments throughout the research process, which significantly contributed to the improvement of this paper. Additionally, we appreciate the valuable comments and suggestions provided by the reviewers and editors.

## Author Contributions

**Conceptualization:** Yan Gong, Yu Qingxu.

**Data curation:** Bowen Zhang, Yu Qingxu.

**Formal analysis:** Bowen Zhang, Yu Qingxu.

**Funding acquisition:** Qinghui Lai, Xiao Chen.

**Investigation:** Yu Qingxu.

**Methodology:** Yu Qingxu.

**Project administration:** Yu Qingxu.

**Resources:** Qinghui Lai, Yan Gong, Yu Qingxu.

**Software:** Yu Qingxu.

**Supervision:** Yu Qingxu.

**Validation:** Yan Gong, Yu Qingxu.

**Visualization:** Yu Qingxu.

**Writing – original draft:** Yuan Wan.

**Writing – review & editing:** Yu Qingxu.

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
