## [Decision Letter · Decision Letter 0]

15 Jan 2024

PONE-D-23-40437Determination of melon seed physical parameters and calibration of discrete element simulation parametersPLOS ONE

Dear Dr. Qingxu,

Thank you for submitting your manuscript to PLOS ONE. After careful consideration, we feel that it has merit but does not fully meet PLOS ONE’s publication criteria as it currently stands. Therefore, we invite you to submit a revised version of the manuscript that addresses the points raised during the review process.

We look forward to receiving your revised manuscript.

Kind regards,

Waqas Saleem, Ph.D

Academic Editor

PLOS ONE

 [This paper was supported by the National Natural Science Foundation of China (Grant No.31960366)].  

5. Please upload a copy of Figure 10, to which you refer in your text on page 23. If the figure is no longer to be included as part of the submission please remove all reference to it within the text.

6. We note that Figure(s) 2, 5, 6, 7, 8, 9, and 11 in your submission contain copyrighted images. All PLOS content is published under the Creative Commons Attribution License (CC BY 4.0), which means that the manuscript, images, and Supporting Information files will be freely available online, and any third party is permitted to access, download, copy, distribute, and use these materials in any way, even commercially, with proper attribution. For more information, see our copyright guidelines: http://journals.plos.org/plosone/s/licenses-and-copyright.

a. You may seek permission from the original copyright holder of Figure(s) 2, 5, 6, 7, 8, 9, and 11 to publish the content specifically under the CC BY 4.0 license. 

Reviewers' comments:

Reviewer's Responses to Questions

**Comments to the Author**

1. Is the manuscript technically sound, and do the data support the conclusions?

Reviewer #1: Yes

Reviewer #2: Yes

2. Has the statistical analysis been performed appropriately and rigorously? 

Reviewer #1: Yes

Reviewer #2: Yes

3. Have the authors made all data underlying the findings in their manuscript fully available?

Reviewer #1: Yes

Reviewer #2: Yes

4. Is the manuscript presented in an intelligible fashion and written in standard English?

Reviewer #1: Yes

Reviewer #2: Yes

5. Review Comments to the Author

Reviewer #1: The paper presents a comprehensive study aimed at improving the accuracy of the discrete element model for Hami melon seeds. The calibration of parameters involved a combination of practical experiments and simulation tests, focusing on various physical characteristics such as triaxial size, mass, moisture content, density, Young's modulus, and more. The authors employed rigorous testing methodologies, including the Plackett-Burman test, steepest climbing test, and quadratic regression orthogonal rotation combination test, to identify significant factors influencing the angle of repose.

Here are some specific comments and suggestions for improvement:

1. In the section 2, on intrinsic parameters, discussed the mechanical properties of cantaloupe seeds. How do these properties impact the design and functionality of seed-metering devices?

2. In the section 2.2, on bench testing, discussed the measurement of collision recovery coefficients and static friction coefficients. How do these coefficients affect the interaction of cantaloupe seeds with the sowing machinery, and what implications do they have for real-world applications?

3. In the section 2.2.3, Describe the process and purpose of the angle of repose determination, and why is it considered a critical parameter in discrete element simulation?

4. In section 2.5, The optimization module in Design-Expert 10.0 was employed for solving regression equations. kindly elaborate on the specific criteria and considerations used for optimization, and how these optimized parameters contribute to the accuracy of the discrete element model?

5. In section 2.6, What are the implications of the relative error in the static seed drop rest angle and dynamic rolling stacking inclination for cantaloupe seeds, and how closely did the simulation results match the actual experiments?

6. How did the authors ensure the reliability of the physical tests for determining parameters such as triaxial dimensions, thousand-seed mass, and moisture content?

7. In section 3, line 507, briefly mentioned the levitation speed of cantaloupe seeds. Kindly discuss the relevance of this parameter in the context of cantaloupe seeding and how it aligns with the overall goals of the study?

8. In section 3, What factors influence the relative error in the angle of repose, and how were these factors determined in the simulation experiments?

9. What are the implications of the relative error in the static seed drop rest angle and dynamic rolling stacking inclination for cantaloupe seeds, and how closely did the simulation results match the actual experiments?

10. Provide more references to support the information presented throughout the manuscript.

11. Proofread the manuscript for grammar, punctuation, and formatting errors. Ensure that the text is clear, concise, and free from typos.

Addressing these comments and making the suggested improvements will make the paper more comprehensive, coherent, and reader friendly.

Reviewer #2: This article describes the simulation of Hami melon seeds and calibration of the micro parameters with DEM. The authors initially obtained the basic parameters of these seeds. Then, the Plackett-Burman (PB) method was employed to screen these parameters. The authors found that the recovery coefficient, static friction coefficient, and rolling friction coefficient of Hami melon seeds significantly affected the angle of repose. The manuscript presents an effective and intriguing study on modeling and calibration of Hami melon seeds, which is suitable for publication in "PLOS ONE." However, to enhance reader comprehension, certain explanations are needed. The following are some highlighted issues.

6. PLOS authors have the option to publish the peer review history of their article (what does this mean?). If published, this will include your full peer review and any attached files.

Reviewer #1: **Yes: **Surinder Pal

Reviewer #2: No

---

## [Author Response · Author response to Decision Letter 0]

23 Feb 2024

Dear editor and reviewers,

Thank you for offering us an opportunity to improve the quality of our submitted manuscript (Determination of melon seed physical parameters and calibration of discrete element simulation parameters). We appreciated the viewers’ constructive and insightful comments. In this revision, we have addressed all of these comments and suggestions. We hope the revised manuscript has now met the publication standards of your journal. 

We highlighted the revisions in red. On the next page, our point-to-point responses to the queries raised by the reviewers are listed. Responds to the reviewer’s comments.

Reviewer#1：

Comment 1: In the section 2, on intrinsic parameters, discussed the mechanical properties of cantaloupe seeds. How do these properties impact the design and functionality of seed-metering devices? 

Response: Physical properties such as seed triaxial dimensions, moisture content, 100-grain weight, density, Poisson's ratio, modulus of elasticity, and shear modulus can have an impact on seed dispenser design and function. The following is a list of some, but not all, of the ways in which these properties can affect the design and performance of a seed dispenser.

Triaxial Dimensions:

They affect the design of the dimensions of the slots and holes in the seed displacer to accommodate seeds of different sizes and shapes. They may affect the flow and positioning of seed within the seed discharger.

Moisture content:

It affects the weight and volume of the seed, which in turn affects the calibration and adjustment of the seed discharger. It may result in adhesion between seeds, requiring adjustment of the seed expeller design to prevent clogging or jamming.

Hundred-grain weight:

It affects the setting and adjustment of the seed discharger to ensure that the required number of seeds are discharged at a time. Adjustments to the speed and vibration of the seed discharger may be required to accommodate seeds of different hundred-grain weights.

Density:

It affects the choice of construction materials and the strength of the design of the seed discharger. It may affect the fluidity of the seed and may require adjustments to the construction of the seed discharger to ensure smooth seed discharge.

Poisson's ratio, modulus of elasticity, and shear modulus:

They affect the structural design and durability of the seed discharger. Adjustments to the elastic elements of seed dischargers may be required to ensure proper seed discharge and to accommodate irregularities in the soil surface.

When designing seed dischargers, these factors need to be taken into account and adjusted to the physical properties of the seed to ensure that the seed discharger works effectively and consistently under different operating conditions. However, the paper did not specify this aspect, as it is relatively basic. Providing too much detail on this topic would make the paper lengthy and cumbersome.

Comment 2: In the section 2.2, on bench testing, discussed the measurement of collision recovery coefficients and static friction coefficients. How do these coefficients affect the interaction of cantaloupe seeds with the sowing machinery, and what implications do they have for real-world applications?

Response: The recovery coefficient for seed collisions and the static friction coefficient are fundamental particle properties essential for conducting discrete element simulations. The precision of the output results in DEM simulations is intricately linked to the accuracy of the input parameters. Simultaneously, in the seed discharger design, elements like the static friction coefficient and seed collision recovery coefficient play a significant role in influencing the seed flow within the discharger, the energy consumption associated with it, and various aspects of mechanical wear on the discharger. In the application process, it is important to select the appropriate material to ensure the collision recovery coefficient and static friction coefficient are suitable. This will increase seed mobility and prevent issues such as seed jamming, filling difficulties, and seed damage. In the Revised Manuscript, 259 lines and 323lines added part of the content for explanation.

Comment 3: In the section 2.2.3, Describe the process and purpose of the angle of repose determination, and why is it considered a critical parameter in discrete element simulation?

Response: The angle of repose reflects the internal friction characteristics and scattering performance of bulk materials. It is an inherent property of materials and is closely related to several influencing factors, such as contact materials, material shape, material size, material moisture content, and the environment in which the materials are stacked. By adjusting the factors that influence the angle of repose, the simulation can more accurately reflect real-world conditions, thereby increasing its reliability and accuracy. In Section 2.2.3, line 353 of the Revised Manuscript, we have revised this portion of the article to clarify the process and significance of determining the angle of repose. We have also explained why the angle of repose is a crucial parameter in discrete element simulations.

Comment 4: In section 2.5, The optimization module in Design-Expert 10.0 was employed for solving regression equations. kindly elaborate on the specific criteria and considerations used for optimization, and how these optimized parameters contribute to the accuracy of the discrete element model?

Response: During the optimization of Design-Expert 10.0, we utilized the software's optimization module to efficiently find the global optimal solution. We relied on three specific criteria:

Minimization of the sum of squares of errors, which aims to minimize the difference between the model-predicted values and the actual observed values, By minimizing the sum of squared errors, it is possible to find parameter combinations that improve the accuracy of model predictions.

It is important to maximize the R-squared value. Additionally, maximizing the R-squared value, which measures the extent to which the model explains the variation in the data, can help the model better capture the relationship between the independent variables.

It is important to consider a range of parameter values. During the optimization process, a range of reasonable values is considered for each parameter. This approach helps to avoid unrealistic parameter values and ensures that the parameter combinations found are feasible in practical applications. This paper aims to minimize the relative error Y in the seed rest angle. To achieve this, we consider the values of collision recovery coefficient, static friction coefficient, and rolling friction coefficients X1, X2, and X3 between cantaloupe seeds. The range of values for these coefficients is 0.47–0.55, 0.54–0.64, and 0.3–0.4, respectively.

By optimizing these parameters, the accuracy of the discrete element model can be improved. This enhances the model's generalization ability and prediction accuracy.

Comment 5: In section 2.6, What are the implications of the relative error in the static seed drop rest angle and dynamic rolling stacking inclination for cantaloupe seeds, and how closely did the simulation results match the actual experiments?

Response: The angle of repose of cantaloupe seeds was compared between the validation results of the static seed drop test and the actual test. In the actual test, the angle of repose was 31.39°, while in the simulation test, it was 31.93°, resulting in a 1.71% higher simulation result compared to the actual test. The validation results of the dynamic rolling stacking test were compared. The dynamic rolling stacking angle of cantaloupe seeds in the actual test was 51°, while in the simulation test it was 51.98°, resulting in a 1.92% higher simulation result than the actual test. This indicates the good accuracy of the discrete element model, providing a reliable basis for subsequent simulation studies of the seed discharger. A part of the modifications has been made in Section 2.6.

Comment 6: How did the authors ensure the reliability of the physical tests for determining parameters such as triaxial dimensions, thousand-seed mass, and moisture content?

Response: We used appropriate measuring tools for various physical parameters of cantaloupe seeds and correctly used various measuring instruments, including but not limited to calipers, the MA-150 infrared moisture detector, the LQ-C50002 electronic balance, the TA.XTPlu professional food physical property analyzer, etc. At the same time, the final data were averaged after multiple sample measurements to minimize measurement errors. For example, in lines 125–128 of the Revised Manuscript, when measuring the triaxial size of cantaloupe seeds, we used vernier calipers (accuracy 0.01 mm) to measure the triaxial size of 500 cantaloupe seeds, and the results were statistically analyzed as shown in Figure 1.

Comment 7: In section 3, line 507, briefly mentioned the levitation speed of cantaloupe seeds. Kindly discuss the relevance of this parameter in the context of cantaloupe seeding and how it aligns with the overall goals of the study?

Response: The suspension velocity of cantaloupe seeds was not considered in the research content of this paper for the time being, and this part has now been removed (line 550 of the Revised Manuscript).

Comment 8: In section 3, What factors influence the relative error in the angle of repose, and how were these factors determined in the simulation experiments?

Response: In both the Abstract and Chapter 3 Results and Discussion sections of this paper, the relative error in angle of repose is affected by three factors in order: the static friction coefficient, the collision recovery coefficient, and the rolling friction coefficient. This conclusion was drawn from the steepest climb test and the quadratic regression orthogonal rotational combination test.

Comment 9: What are the implications of the relative error in the static seed drop rest angle and dynamic rolling stacking inclination for cantaloupe seeds, and how closely did the simulation results match the actual experiments?

Response: Suggestion #9, which you provided, is identical to suggestion #5, as amended and explained on line 529-539 of the Revised Manuscript.

Comment 10: Provide more references to support the information presented throughout the manuscript.

Response: References [22, 23] have been added at line 259, and reference [25] has been added at line 323 to support the information provided in the Revised Manuscript. 

Comment 11: Proofread the manuscript for grammar, punctuation, and formatting errors. Ensure that the text is clear, concise, and free from typos.

Response: We have proofread the article in response to your request to improve its semantic coherence.

Reviewer#2：

Comment 1: Why was the inclinometer's plate replaced with an ABS plastic plate for testing? If there are some advantages, it would be beneficial to articulate them in the article.

Response: During seeding research, 3D printing is utilized for seed dispenser fabrication due to its efficient and rapid processing of complex parts, which accelerates the research process. ABS is a commonly used 3D printing material. Therefore, ABS was used as the contact material throughout the calibration tests, and the coefficient of friction determination test was no exception. This part has been added in line 343-351 of the Revised Manuscript to explain the advantages of using ABS material.

Comment 2: How many particles are required to model a seed accurately? Please provide a more detailed explanation.

Response: To ensure similarity of the seed appearance models while taking into account computational accuracy and speed, we randomly selected 300 spherical particles with diameters ranging from 0.1 to 1.19 mm to populate the seed models. This section has been added to line 398-401 of the Revised Manuscript, which explains how many particles are required to accurately simulate a seed.

Comment 3: The x and y axes in Figure 8 should be explained, including their units of measurement.

Response: The corresponding changes were made in Section 2.2, the respective meanings of x and y were explained, and Figure 8c was redrawn. 

Comment 4: There is Chinese text in the P-values section of Table 6. Please make the necessary modifications.

Response: Already revised in the paper.

Thank you very much. According to your comment, we have had the manuscript polished and corrected the mistakes.

Yours sincerely,

Qingxu Yu

February 23, 2024

---

## [Editor Report · Decision Letter 1]

29 Feb 2024

Determination of melon seed physical parameters and calibration of discrete element simulation parameters

PONE-D-23-40437R1

Dear Dr. Qingxu,

We’re pleased to inform you that your manuscript has been judged scientifically suitable for publication and will be formally accepted for publication once it meets all outstanding technical requirements.

Kind regards,

Waqas Saleem, Ph.D

Academic Editor

PLOS ONE

---

## [Editor Report · Acceptance letter]

29 May 2024

PONE-D-23-40437R1 

PLOS ONE

Dear Dr. Qingxu, 

I'm pleased to inform you that your manuscript has been deemed suitable for publication in PLOS ONE. Congratulations! Your manuscript is now being handed over to our production team.

Kind regards, 

on behalf of

Dr. Waqas Saleem 

Academic Editor

PLOS ONE